# Ultrasound-Responsive Liposomes for Targeted Drug Delivery Combined with Focused Ultrasound

**DOI:** 10.3390/pharmaceutics14071314

**Published:** 2022-06-21

**Authors:** Yoon-Seok Kim, Min Jung Ko, Hyungwon Moon, Wonchul Sim, Ae Shin Cho, Gio Gil, Hyun Ryoung Kim

**Affiliations:** R&D Center, IMGT Co., Ltd., 172, Dolma-ro, Bundang-gu, Seongnam-si 13605, Gyeonggi-do, Korea; yoonseok.kim@nanoimgt.com (Y.-S.K.); minjeong.ko@nanoimgt.com (M.J.K.); hyungwon.moon@nanoimgt.com (H.M.); wonchul.sim@nanoimgt.com (W.S.); aeshin.cho@nanoimgt.com (A.S.C.); gio.gil@nanoimgt.com (G.G.)

**Keywords:** ultrasound-responsive liposome, liposomal doxorubicin, stimuli-triggered drug release, high-intensity focused ultrasound, triple-negative breast cancer

## Abstract

Chemotherapeutic drugs are traditionally used for the treatment of cancer. However, chemodrugs generally induce side effects and decrease anticancer effects due to indiscriminate diffusion and poor drug delivery. To overcome these limitations of chemotherapy, in this study, ultrasound-responsive liposomes were fabricated and used as drug carriers for delivering the anticancer drug doxorubicin, which was able to induce cancer cell death. The ultrasound-sensitive liposome demonstrated a size distribution of 81.94 nm, and the entrapment efficiency of doxorubicin was 97.1 ± 1.44%. The release of doxorubicin under the ultrasound irradiation was 60% on continuous wave and 50% by optimizing the focused ultrasound conditions. In vivo fluorescence live imaging was used to visualize the doxorubicin release in the MDA-MB-231 xenografted mouse, and it was demonstrated that liposomal drugs were released in response to ultrasound irradiation of the tissue. The combination of ultrasound and liposomes suppressed tumor growth over 56% more than liposomes without ultrasound exposure and 98% more than the control group. In conclusion, this study provides a potential alternative for overcoming the previous limitations of chemotherapeutics.

## 1. Introduction

The current strategies for the treatment of cancer involve surgical dissection, radiotherapy, and chemotherapy. Among the means of cancer therapy, chemotherapy is one of the most used methods compared to the other strategies. Conventional chemotherapy induces a toxic effect in tumor cells and a suppression of tumor growth. However, chemotherapeutics also cause side effects in healthy tissues and biological systems due to the random or wide distribution of drugs. To overcome this disadvantage, drug carriers have received great attention for the treatment of tumors [1,2]. Drug-loaded carriers have great potential as a targeting treatment that can increase therapeutic efficacy by increasing the accumulation of nanosized carriers into tumor tissue [3,4]. Most solid tumors have distinct characteristics compared with healthy tissues, such as a leaky vascular architecture caused by rapid angiogenesis, resulting in enhanced permeability of drugs through blood vessels. This phenomenon is known as enhanced permeability and retention (EPR). Nanosized carriers can cross the leaky blood vessels of tumor tissues and accumulate in tumors via the EPR effect [5,6,7].

There are several types of drug carriers for anticancer drugs. Among the nanosized drug carriers, liposomes lead the market because of their unique properties and broad range of biomedical applications [8,9,10]. Among the various nano-sized drug carriers, liposomes are able to stabilize drugs, increase tissue uptake, and improve bioavailability [11,12]. The shape of a liposome is a spherical vesicle with a lipid bilayer enclosing a discrete aqueous core. Liposomes can encapsulate a diverse range of drugs because they have an aqueous core and a hydrophobic lipid bilayer. The encapsulated chemotherapeutics in liposomes are stably protected from physiological events [13]. Liposomes are basically composed by the combination of various biocompatible phospholipids. In addition, the physio-biological properties of liposomes are sufficiently dependent on the types of lipids constituting liposomes. Regarding the lipid composition, the shell properties, such as the surface charge, stiffness, and rigidity, critically affect the pharmacokinetics, biodistribution, and excretion. Therefore, in order to reduce the side effects of chemotherapeutics and to enhance the anticancer effects, shell properties are controlled by changing the species and ratios of lipids [14]. However, conventional liposomes (DOXIL) are still limited for the effective treatment of cancer, despite advances in long circulation and the reduction in side effects. The sufficient stable shells of liposomes lead to a lower release of chemotherapeutics. Therefore, the anticancer effect has been limited by a low accumulation of active chemotherapeutics in tumor cells. Hence, to achieve effective drug release from liposomes, local stimuli-responsive liposomes such as pH-, temperature-, radio frequency radiation-, and ultrasound-responsive liposomes have been intensively researched. Among these energy triggers, the combination with ultrasound was recently highlighted as a non-invasive and conveniently controllable procedure.

Ultrasound has been actively applied for the clinical field, especially high-intensity focused ultrasound, which has been approved for the ablation of uterine fibroids in the human clinic [15,16]. In addition, the application of high-intensity focused ultrasound to other abnormalities, such as cancer and brain diseases, has been continuously expanded by human investigational trials [17,18]. The unique advantage of ultrasound is that it is capable of the target-specific delivery and release of therapeutics by ultrasound pressure. The high intensity of ultrasound pressure can induce an endogenous microbubble, and the microbubble is repeatably expanded and shrunk by its resonant size and then implodes under the ultrasound wave. This behavior is called cavitation creating high gas pressures and temperatures [19]. These energies can induce particle disruption and are suitable for releasing drugs from liposomes and other nanomedicines. Therefore, ultrasound is capable of enhancing the target-specific delivery and selective release of chemotherapeutics dependent on ultrasound pressure [20].

Regarding the advantage of the ultrasound-triggered release of drugs, an ultrasound-sensitive liposome encapsulating doxorubicin (IMP301) was developed in this study. IMP301 was designed for doxorubicin release under the specific condition of focused ultrasound pressure by the composition and ratio of phospholipids. This characteristic is able to not only reduce the side effects of doxorubicin but also enhances the anticancer effect. In this study, the physicochemical characteristics and sono-sensitivity of IMP301 were verified by in vitro experiments with a comparison to conventional liposomal doxorubicin (DOXIL). In addition, the anticancer effect of IMP301 was investigated by its pharmacokinetics, biodistribution, and in vivo efficacy.

## 2. Materials and Method

### 2.1. Materials

1,2-Distearoyl-sn-glycero-3-phosphocholine (DSPC), (*N*-(carbonyl-methoxypolyethylene glycol-2000)-1,2-distearoyl-sn-glycero-3-phosphoethanolamine, and sodium salt) (DSPE-mPEG2000), and 1,2-dioleoyl-sn-glycero-3-phosphoethanolamine (DOPE) were purchased from Lipoid AG (Steinhausen, Switzerland). 1-Stearoyl-2-lyso-sn-glycero-3-phosphocholine (MSPC, S-LysoPC) was purchased from NOF America Corporation (White Plains, NY, USA). Doxorubicin hydrochloride (DOX) was purchased from Gemini Pharmaceuticals Inc., (Hauppauge, NY, USA). Cholesterol, ammonium sulfate, hydrochloric acid, sodium hydroxide, 1-(4,5-Dimethylthiazol-2-yl)-3,5-diphenylformazan, thiazolyl blue formazan (MTT), and L-histidine were purchased from Sigma-Aldrich (St. Louis, MO, USA). Sucrose was purchased from CheilJedang (Seoul, Korea). An Oasis HLB 3 cc Vac Cartridge (solid-phase extraction column; SPE column) was purchased from Waters (Milford, MA, USA).

### 2.2. Preparation of DOX-Loaded Liposomes

The lipid composition of IMP301, which was obtained from a pilot study, was DSPC/DSPE-PEG/cholesterol/DOPE/MSPC. IMP301 was fabricated by ethanol injection followed by extrusion. Briefly, 1.50 g of DSPC, 2.66 g of DSPE-PEG, 2.20 g of cholesterol, 9.16 g of DOPE, and 0.50 g of MSPC were dissolved in 62.5 mL of ethanol. The organic phase was gently heated to 60 °C to dissolve the lipid components. Then, the lipid-containing ethanol was injected into 437.5 mL of 250 mM ammonium sulfate solution at 250 rpm. Multilamellar vesicles were assembled and dispersed during ethanol injection and downsized by serial extrusion cycles with polycarbonate filter pore sizes ranging from 200 to 80 nm using a LIPEX^®^ 800 mL Thermobarrel extruder (Evonik, BBY, Burnaby, BC, Canada). The temperature of the vesicles was maintained at 50 °C during the extrusion. The dispersion of extruded liposomes was exchanged with pH 6.5 10% sucrose and a 10 mM histidine buffer using a 12–14 kDa dialysis membrane. The ammonium gradient across the liposomal membrane was generated by exchanging ammonium sulfate with the buffer. DOX was encapsulated into the intraliposomal aqueous phase using the remote loading method. DOX was added to the liposome dispersion at a ratio of 1:8 to liposomes and stirred at 37 °C for 2 h. DOX-loaded liposomes were diluted with a buffer solution, so that the DOX concentration was 2 mg/mL, and stored at 2–8 °C.

### 2.3. Characterization of IMP301

The size distribution of IMP301 was measured using dynamic light scattering (DLS; Nano ZS90, Malvern Panalytical, Malvern, UK). The osmolarity of IMP301 was measured using an osmometer (OsmoTECH^®^ XT, Advanced Instruments, Norwood, MA, USA). The temperature was maintained at 25 °C. DOX and lipids were quantified by a high-performance liquid chromatography (HPLC; 1260 Infinity II, Agilent Technologies, Santa Clara, CA, USA) system with a diode array detector and an evaporative light scattering detector. The extraction of free DOX for the measurement of the encapsulation efficacy was carried out using the same method as that used for the quantification of released DOX. The morphology of liposomes was examined using cryo-transmission electron microscopy (cryo-TEM). Specimens for cryo-TEM were prepared using an FEI Vitrobot-type FP5350/60. A liposome solution (5 µL) was applied to a carbon TEM grid (Lacey support films, 200 mesh, Ted Pella, Inc., Redding, CA, USA) blotted with filter paper and plunged into liquid ethane. In addition, the encapsulation efficiency of doxorubicin was analyzed with a UV-vis system after purifying the unloaded DOX using a PD-10 column. The quantification of doxorubicin was measured by the absorbance at 475 nm.

### 2.4. Analysis of In Vitro Release under Ultrasound Exposure

To conduct the DOX release test of IMP301, a plane wave ultrasound was irradiated by a pin-type ultrasonicator for 1 min at a frequency of 24 kHz and a continuous ultrasound wave intensity of 92 kW/cm^2^. Briefly, 2 mL of liposomal suspension ultrasound irradiated with a pin sonicator was loaded onto a desalting column, followed by the addition of distilled water (0.5 mL). An additional 4 mL of distilled water was added to the desalting column and collected in a cuvette to measure the absorbance of liposomal DOX at 475 nm. A reduced absorbance compared with non-irradiated liposomes indicated the amount of DOX released. The percentage of released drug was calculated using the following equation:$$\%\;\mathrm{release}=(1-\frac{A_{r}}{A_{o}}\;)\;\times \;100$$
where *A_r_* and *A_o_* represent the absorbance intensities of the sonicated liposome suspension and of the original one, respectively.

### 2.5. Determination of the Ideal FUS Parameters for DOX Release

To analyze the effect of FUS on DOX release, focused ultrasound (FUS) machinery (VIFU 2000, ALPINION, Anyang-si, Korea) was used with a water bath equipped with a temperature controller and degasser. The center frequency of the FUS transducer was 1 MHz, and the beam resolution of the focal zone at 6 dB demonstrated a circular shape with a 1 mm diameter circle and a length of 1 cm. To investigate FUS-triggered DOX release, IMP301 was exposed to FUS with varying FUS parameters of intensity, duty cycle, pulse repetition frequency (PRF), and exposure time/spot. Prior to FUS irradiation, IMP301 was contained in the dialysis membrane and was kept at 37 °C by degassing water for 1 h. The quantification of DOX release was analyzed by measuring the optical absorbance at 475 nm. After FUS irradiation, the released DOX was purified using a desalting column, and its amount was calculated as described above. Commercial liposomal doxorubicin, DOXIL, was used as an ultrasound-insensitive liposome.

### 2.6. IMP301 Stability Test

Stability tests were carried out in 10% sucrose and 10 mM histidine buffer solutions at a pH of 6.5. IMP301 was filtered through a 0.2 µm bottle top filter to prevent germ growth caused by the dispersant. Then, 10 mL of IMP301 (2 mg/mL) was packed in a sterilized 20 mL vial with aluminum seals. Aliquots of IMP301 were stored at 2–8 °C in the dark. To assess stability, each vial was used for measuring the DOX content, entrapment, lipid content, size, and polydispersity index (PDI) of IMP301 at 0, 1, 3, 6, and 9 months. The amount of DOX released was analyzed with an HPLC system after extracting the released DOX using a serum-coated SPE column. The extraction of DOX from the SPE column was performed using a 1:1 mixture of distilled water and acetonitrile. The concentration of DOX in the collected solution was obtained by measuring the absorbance at 254 nm in an HPLC system equipped with a C-18 reverse phase column. The phospholipids were quantified using an evaporative laser scattering detector.

### 2.7. In Vitro Cellular Uptake and Cytotoxicity of IMP301

A cellular uptake study was performed using MDA-MB-231 human breast cancer cells. Three groups treated with IMP301, DOXIL (Batch No. LBZSC00), and free DOX were compared. MDA-MB-231 cells were cultured in Dulbecco’s modified Eagle’s medium (DMEM) supplemented with 10% fetal bovine serum and 1% antibiotic–antimycotic. MDA-MB-231 cells were seeded at a density of 2.5 × 10^4^ cells/well 24 h prior to treatment on 8-well glass chamber slides with culture medium. IMP301, DOXIL, and free DOX were added to each well for 30 min. To compare the efficiency of DOX uptake, exposure to each agent was carried out for 1 min at low-frequency ultrasounds (24 kHz). MDA-MB-231 cells were washed with phosphate-buffered saline (PBS; 0.01M, pH 7.4) and fixed with 4% paraformaldehyde for 3 min. The cellular uptake was observed using a confocal laser scanning microscope. 

The MDA-MB-231 cell line was used for the in vitro cytotoxicity test. Cells (1 × 10^4^) were seeded in a 96-well plate. After overnight incubation, 2, 4, 6, 8, and 10 µg/mL IMP301 and DOXIL were applied to each well. Ultrasound-treated particles were prepared using the same method used for the cellular uptake investigation. Cells were incubated with a medium containing the drugs for 3 h, which was then replaced with fresh culture medium after washing three times with PBS. After 72 h of incubation, the MTT assay was conducted to evaluate cell viability.

### 2.8. Pharmacokinetics of Ultrasound-Sensitive Liposomal DOX

All animal experiments were conducted according to the guidelines of the Institutional Animal Care and Use Committee (IACUC) of Seoul National University Bundang Hospital (approval number: MSRI-51-19-011, 28 April 2020). ICR mice (6 weeks old) were prepared for the pharmacokinetic studies. Mice were grouped into the free-DOX and IMP301 groups and injected intravenously with the respective drug preparation. Blood samples were collected at different time points (2, 5, 15, 30, 60, 120, 240, 480, 1440, and 2880 min). DOX was quantitatively measured in blood plasma. Briefly, mouse plasma mixed with DOX solution was prepared as a standard solution, which was mixed with daunorubicin as an internal standard, followed by centrifugation at 13,000 rpm. The supernatant was diluted with a 30% acetonitrile solution containing 0.1% formic acid. Quantitative measurements of DOX were performed using LC-MS/MS. The DOX calibration curve was plotted using the DOX and daunorubicin area ratio. The DOX concentration in mouse plasma was measured after dilution with PBS containing 20% methanol and 5% BSA.

### 2.9. In Vivo DOX Release under Ultrasound Irradiation

A subcutaneous MDA-MB-231 human breast cancer xenograft model (female, balb/c nude mice, 12~14 weeks old) was used to evaluate DOX release to the tumor region under FUS irradiation using in-house-developed FUS machinery, IMD-10R. IMP301 and DOXIL (20 mg/kg) were administered intravenously. FUS with a 2.0 kW/cm^2^ intensity, 2.0% duty cycle, 250 Hz PRF, and 10s/spot was irradiated after each agent administration. To quantify the release of DOX in tumors, an in vivo fluorescence imaging system was used to visualize the fluorescence images of the mice. The fluorescence intensity was mainly detected at λ_Ex_/λ_Em_ = 470/560 nm. To eliminate the auto-fluorescence biomolecules from the mouse body, spectral unmixing was conducted, and only the fluorescence intensity of DOX was observed.

### 2.10. In Vivo Anticancer Efficacy

MDA-MB-231 xenografted mice (*n* = 3) were prepared for in vivo anticancer efficacy experiments. MDA-ME-231 cells were inoculated at 5 × 10^6^ per mouse. To assess the synergistic effect of IMP301 and ultrasounds, nine mice were randomly divided into three groups: (i) control, (ii) IMP301 (IMP301, US-), and (iii) IMP301 under FUS irradiation using VIFU 2000 (IMP301, US+). Saline was injected in the control group (i). FUS was irradiated with the condition of 2.8 kW/cm2, 30 s/spot, PRF 250 Hz, and duty cycle 5%. All groups were treated with the drugs on days 1, 4, and 8. The dose of doxorubicin was administrated at 2 mg/kg. In addition, the FUS exposure group was irradiated with the ultrasound at the tumor region after 1 h of administration. To investigate the tumor suppression efficiency, tumor volume and body weight were measured every 3 days for 1 month. The tumor volume was calculated using the equation 0.5 × short length × long length × height, and the distributions of DOX, DOXIL, and IMP301 to the heart were quantitively analyzed by LC-MS/MS.

## 3. Results and Discussion

### 3.1. Characterization of IMP301

We developed an ultrasound-sensitive liposomal DOX, IMP301, which formed an unclear red suspension. The liposomes showed a spherical shape, an average diameter of 81.94 nm, and a PDI of 0.087, indicating a narrow size distribution (Figure 1a). The DOX absorbance peak appeared after DOX loading in the absorbance spectrum, indicating successful DOX encapsulation with a loading efficiency of 97.1 ± 1.44% (Figure 1b). Meanwhile, DOX crystalized by ion exchange originated rod-shaped liposomes (Figure 1c). The efficiency of DOX release increased depending on the irradiation time, and IMP301 showed a higher DOX release efficiency than DOXIL (Figure 1d). The osmolarity and pH of IMP301 were 320 mOsm/kg and 6.5, respectively. IMP301 was shown to be stable for at least 9 months when stored at 2–8 °C. The contents of DSPC, DSPE-mPEG2000, cholesterol, DOPE, MSPC, and DOX were maintained at 108.7%, 106.5%, 101.5%, 109.0%, 99.2%, and 97.5%, respectively, for 9 months.

### 3.2. In Vitro Release of DOX under the FUS Exposure

To compare the sonosensitivity of IMP301 and DOXIL, both liposomes was exposed to continuous ultrasound waves, and the release amount of doxorubicin was quantitatively measured. Under ultrasound irradiation with a 92 W/cm^2^ intensity at 24 kHz, IMP301 demonstrated a DOX release ratio of 58.8%. In contrast, that of DOXIL was only 10.2% in equal condition of ultrasound exposure. The IMP301 released doxorubicin more than five times more than DOXIL. Basically, the characteristics of the liposomes were sufficiently related to the composition and ratio of the phospholipid types. The phospholipid structure determines the thermodynamics and physicochemical properties of liposomes. In the case of IMP301, the surface of IMP301 was rearranged to the inverted-cone-shape under ultrasound stimuli due to the DOPE structure. DOPE is composed of a small hydrodynamic volume of hydrophilic heads and a large one of hydrophobic tails, defining a high packing parameter (PP > 1) and seems to induce destabilization by converting lamellar to the inverted hexagonal phase under acoustic wave pressure [21]. Conversely, DOXIL predominantly consists of HSPC, demonstrating a cylindrical structure with an equal volume of hydrophobic chains and hydrophilic heads. This structure aligns the linear bilayer of the liposome shell and increases the stability of the liposome. Therefore, IMP301 significantly increased DOX release under ultrasound pressure compared to DOXIL. In the FUS experiment irradiating pulse wave, the DOX release ratio was similar to that of the continuous ultrasound wave. The DOX release ratios of IMP301 and DOXIL were 48.1% and 23.3%, respectively (Figure 2a). The ultrasound-induced doxorubicin release of IMP301 was confirmed using cryo-TEM images. The shape of the liposomes changed after ultrasonic irradiation. The thickness of the DOX crystal rod in the liposomes decreased after ultrasound exposure. The broad size distribution range also showed the ultrasound-induced destabilization of liposomes (Figure 2b,c).

### 3.3. In Vitro Release Depends on FUS Parameters

To explore the optimal ultrasound conditions for in vivo use, drug release experiments were conducted under different ultrasound conditions (intensity, PRF, irradiation time, and duty cycle). The DOX release tended to increase as a function of ultrasound strength. To effectively release DOX from the inner liposome, a specific intensity threshold is required; in the case of IMP301, at least 2.8 kW/cm^2^ is required (Figure 3a). In particular, PRF significantly affected the release behavior of IMP301. The drug release rate tended to increase with a PRF up to 250 Hz, but it decreased above 250 Hz of PRF (Figure 3b). PRF is a critical factor related to cavitation generation [22]. Because the frequency is related to the wave pressure of the medium, an increase in the PRF induced strong shock waves and enhanced the release of DOX from IMP301. The irradiation time also affected the DOX release in IMP301 (the release rate increased according to the irradiation time; Figure 3c). Furthermore, to investigate an important factor for the release of DOX from IMP301, the ultrasound was irradiated with varying duty cycles and intensities. IMP301 was equally exposed to the total energy of the ultrasound with a spatial-peak temporal average (ISPTA) of 56 W/cm^2^ (Figure 3d). The duty cycle and intensity are related to heat and pressure. Drug release occurred at intensities of 2.8 kW/cm^2^ or higher. The frequent application of low-intensity energy slightly affected drug release, but the temporary application of high-intensity energy induced the release of a larger amount of drug, even when irradiated with the same amount of energy. Therefore, the mechanism of doxorubicin release form IMP301 was dependent on the FUS intensity, not on temperature (Appendix A). For in vivo applications, the use of low-intensity ultrasound is essential to avoid tissue damage. The ultrasonic conditions for the animal experiment were set at 2.8 kW/cm^2^ of intensity, 250 Hz of PRF, 10 s exposure time, and a 2% duty cycle.

### 3.4. Long-Term Stability of IMP301

To evaluate the stability of IMP301, the size distribution, phospholipid amount, and entrapment efficiency were measured for 9 months at several time points (0, 1, 3, 6, and 9 months). The starting masses of DOX and all phospholipids for liposome production were standard. The standard ranges of DOX content, lipid content, and entrapment efficiency were set as 90–110% and verified by the doxorubicin release ratio under the ultrasound exposure (data not shown). The results showed a range of 90–110% in the initial month, and the IMP301 composition remained consistently in this standard range of 90–110% for 1, 3, 6, and 9 months (Table 1). The appearance of IMP301 was an unclear red suspension at all time points, which corresponded to a stable lipid components content. Theoretically, phospholipids are easily hydrolyzed in the aqueous phase, originating impurities such as stearic acid, oleic acid, phosphocholine, and phosphatidylethanolamine. The most representative impurity of IMP301 was stearic acid, which was derived from DSPC, DSPE-mPEG2000, and MSPC. However, the total percentage of impurities, including stearic acid, was less than 1% (data not shown). Therefore, DOX was robustly encapsulated for nine months without a considerable number of impurities. The size of the IMP301 liposomes is also an important factor for biodistribution and pharmacokinetics. However, the size distribution and polydispersity index did not significantly change after nine months. Overall, IMP301 was found to be stable for up to nine months, but further stability tests are required to measure its long-term stability.

### 3.5. Cellular Uptake of Liposomes

To investigate the effect of the combination of IMP301 and ultrasound, a cellular uptake study was performed using confocal microscopy and MDA-MB-231 cells. Nuclei were stained with DAPI (blue fluorescence), and red fluorescence showed DOX uptake and localization in cell nuclei. In the ultrasound-exposure IMP301 group, the fluorescence intensity by DOX was co-localized to the DAPI-stained area (Figure 4). This result indicated that the DOX released by ultrasound was accumulated in the nucleus. Basically, doxorubicin is a chemotherapeutic that intercalates DNA and induces anticancer effects by the protection of DNA duplication. Therefore, DOX released from IMP301 accumulated to the nucleus and indicated effective cytotoxicity under ultrasound exposure. On the other hand, with OXIL, the red fluorescence released by DOX in the nucleus was not observed, even in the group irradiated with ultrasound. DOXIL is sufficiently stable, even under ultrasound exposure. In addition, DOXIL was not capable of penetrating into the MDA-MB-231 cell line. Therefore, Figure 4 showed that IMP301 is capable of an ultrasound-triggered release of DOX.

### 3.6. Cytotoxicity of IMP301 with Ultrasound-Triggered DOX Release

To verify ultrasound-triggered cell cytotoxicity, an in vitro MTT assay was carried out using the MDA-MB-231 cell line. Ultrasound-insensitive liposomal DOX, DOXIL, was used for comparison with IMP301. Each drug was divided into groups irradiated with ultrasound and a non-irradiated group. All groups were treated with cells at concentrations ranging from 2 to 10 µg/mL (Figure 5). In the case of non-exposure to ultrasound, there were no significant differences between IMP301 and DOXIL across all drug concentration ranges. As shown in Figure 4, MDA-MB-231 cells were not readily permeable to the DOX of IMP301 or DOXIL without ultrasound. Therefore, cellular uptake and the related anticancer effect were relatively low. However, ultrasound-pre-treated IMP301 showed concentration-dependent cytotoxicity, whereas the cytotoxicity of DOXIL was not enhanced compared to the untreated group. Therefore, IMP301 at 6 µg/mL suppressed MDA-MB-231 cell viability at over 50% (45.17 ± 5.15%) On the other hand, DOXIL (101.83 ± 9.61%) was not suppressed at an equal concentration. In addition, free DOX demonstrated similar cytotoxicity both the US- and US+ groups.

### 3.7. Pharmacokinetic Study

To study the pharmacokinetics, IMP301 and free DOX (fDOX) were administered via the intravenous route. The concentration of DOX in the plasma was assessed using LC-MS/MS. A rapid reduction in the DOX concentration in the plasma was observed in the free DOX (fDOX) group. The results in this group confirmed that DOX showed rapid clearance, and small amounts of DOX were measured at all time points. However, the IMP301 treatment group showed a relatively higher concentration of DOX in the plasma than the fDOX treatment group (Figure 6). It was confirmed that liposomes could increase the drug circulation time and inhibit clearance from the plasma. The area under the plasma level–time curve of each group at 24 h post-injection was 666.1 h × ng/mL and 258,995.7 h × ng/mL. IMP301 was higher than fDOX through the entire curve. The C_max_ values of the fDOX- and IMP301-treated groups were 7978.6 and 80,824.7 ng/mL, respectively. As DOX showed low levels of cytotoxicity in the in vitro test, it is expected that there will be little concern, even if the blood concentration is high.

### 3.8. In Vivo Release Test of IMP301 at the Tumor

IMP301 and DOXIL ultrasound-triggered release were compared using live fluorescence imaging analysis. The DOX fluorescence emission of IMP301 and DOXIL was quenched in the core of both liposomes, whereas the fluorescence intensity of DOX was considerably enhanced by its release [23]. Therefore, IMP301 and DOXIL were intravenously administered to MDA-MB-231 xenografted mice, and the tumor region was immediately irradiated with FUS. Figure 7c shows DOX release after FUS exposure in in vivo experiments. IMP301 was slightly released in the tumor region without FUS exposure. Thus, DOX was strongly released from IMP301 after 1 h of FUS irradiation, and the fluorescence intensity increased in a time-dependent manner by the release of DOX. On the other hand, DOXIL was rarely observed in the tumor region in both the FUS-exposure and non-exposure groups. The fluorescence intensity of DOXIL was constantly quenched for 6 h. According to this result, DOXIL was considerably stable, regardless of FUS exposure. Conversely, IMP301 effectively released DOX under FUS irradiation, and it is expected that the combination of FUS and IMP301 enhances the anticancer effects of DOX.

### 3.9. Therapeutic Effect of Ultrasound-Responsive Liposomes In Vivo

None of the groups showed significant body weight loss, indicating low systemic toxicity (Figure 8b). The FUS combined group showed superior tumor growth inhibition efficacy compared to the FUS non-treated group. Tumor growth was observed in the control group. The therapeutic efficacy in the liposomal DOX-treated group was 41% higher than that of the control group. Liposomal DOX exposed to FUS showed nearly 98% more suppression of tumor growth compared with the control. These results indicate a synergistic effect of ultrasound and IMP301 in the treatment of tumors. FUS induced the release of DOX from the liposomes at the tumor site without systemic toxicity. In addition, the distribution of IMP301 to heart was compared to free doxorubicin and DOXIL. The populations of doxorubicin in IMP301 (1419 ± 273 ng/g) and DOXIL (1066 ± 252 ng/g) were about two times lower than free doxorubicin (2616 ± 119 ng/g). Therefore, IMP301 was capable of overcoming the cardiotoxicity of doxorubicin (Appendix A).

## 4. Conclusions

In this study, ultrasound-sensitive liposomal DOX was developed to enhance cancer treatment. IMP301 was uniformly produced with liposomes with diameters of less than 100 nm and a production yield of over 95%. To determine the characteristics of the ultrasound response, in vitro DOX release was analyzed using different FUS parameters (intensity, duty cycle, PRF, and exposure time). The release of DOX was proportional to the total ultrasound energy with respect to the intensity and exposure time. Interestingly, the release of DOX from IMP301 was critically related to the intensity and PRF. For effective DOX release, the intensity is more important than the duty cycle, and PRF for cavitation is essential to generate strong acoustic wave pressure. The release ratio of DOX under FUS irradiation was two times higher than that of DOXIL. The in vivo fluorescence imaging showed that in IMP301 the drug was released from the liposomes in response to FUS. The combination of IMP301 and FUS showed great therapeutic efficacy in MDA-MB-231 tumor-bearing mice. DOX released from liposomes accumulated in tumor tissues and killed cancer cells, leading to tumor suppression.

Ultrasound combination therapy could facilitate the delivery of drugs into specific areas in the body with ultrasound imaging [24,25,26,27,28]. Since a diverse range of drugs can be encapsulated into liposomes, this can be applied to various indications as a platform technology [27,28]. Furthermore, the use of ultrasound-responsive liposomes for targeted drug delivery is a highly interesting area of research with much potential that should be further explored.

## Figures and Tables

**Figure 1 pharmaceutics-14-01314-f001:**
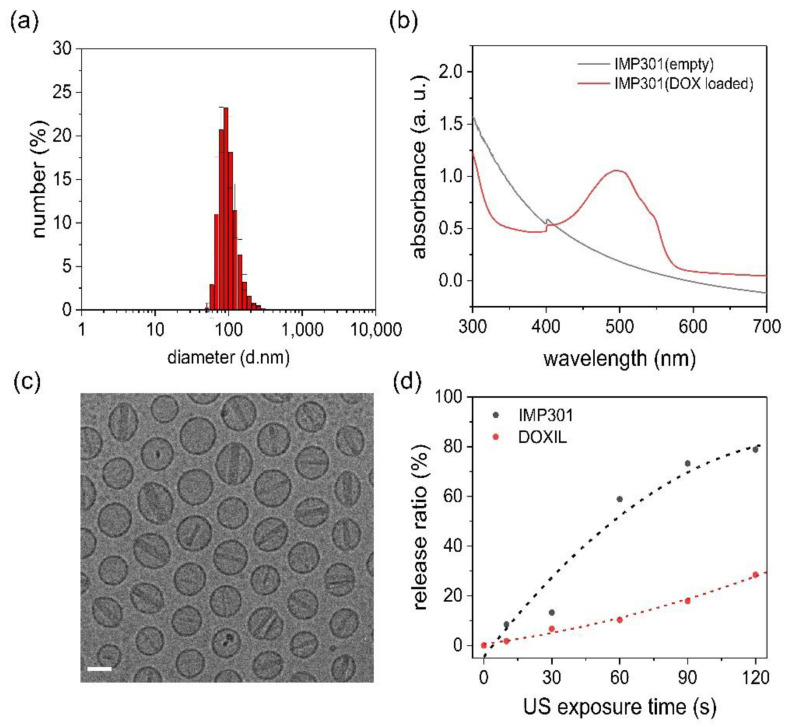
Characterization of IMP301. (**a**) Size distribution of IMP301, (**b**) absorbance spectrum of empty IMP301 (gray line) and doxorubicin (DOX)-loaded IMP301 (red line), (**c**) cryo-TEM image of IMP301 (scale bar = 50 nm), (**d**) the release behavior of IMP301 (black dots) and DOXIL (red dots) depends on ultrasound irradiation time.

**Figure 2 pharmaceutics-14-01314-f002:**
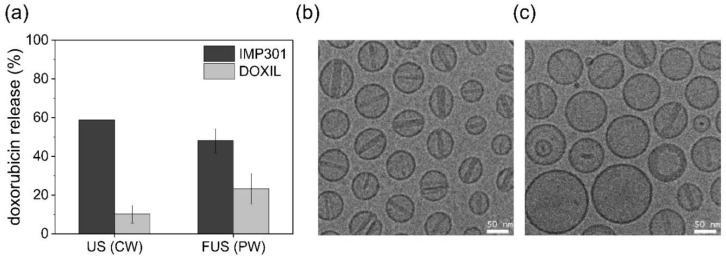
In vitro release behavior of IMP301 and DOXIL. (**a**) Continuous wave ultrasound (CW)- and FUS (pulsed wave, PW)-triggered release of IMP301 (dark gray) and DOXIL (light gray); *n* = 3. TEM image of IMP301 (**b**) before ultrasound irradiation and (**c**) after ultrasound irradiation.

**Figure 3 pharmaceutics-14-01314-f003:**
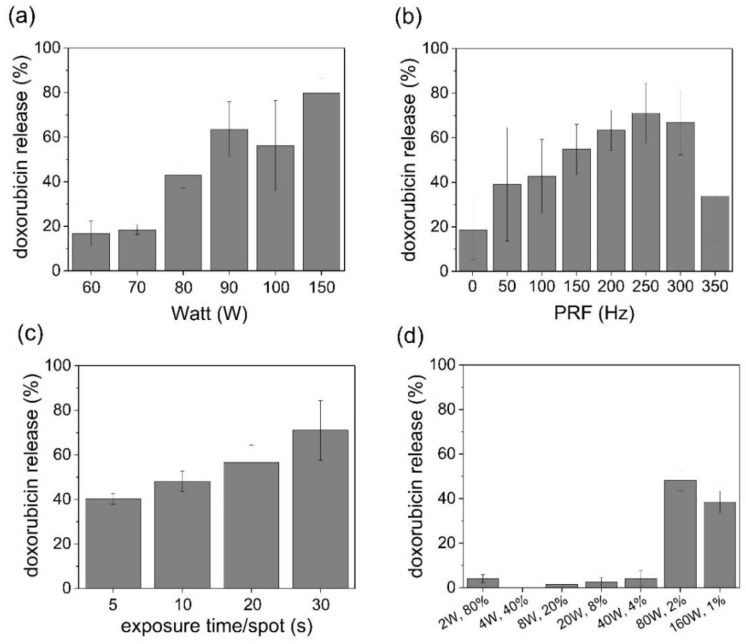
In vitro doxorubicin (DOX) release patterns in IMP301 according to different FUS parameters. (**a**) The amount of DOX released depended on intensity (at 250 Hz PRF, 10 s/spot, and 2% duty cycle). (**b**) The amount of DOX released depended on PRF (at 2.8 kW/cm^2^, 10 s/spot, and 2% duty cycle). (**c**) The amount of DOX released depended on irradiation time (at 2.8 kW/cm^2^, 250 Hz PRF, and 2% duty cycle). (**d**) The amount of DOX released depended on intensity (W) and duty cycle (%) (at 2.8 kW/cm^2^, 250 Hz PRF, and 10 s/spot).

**Figure 4 pharmaceutics-14-01314-f004:**
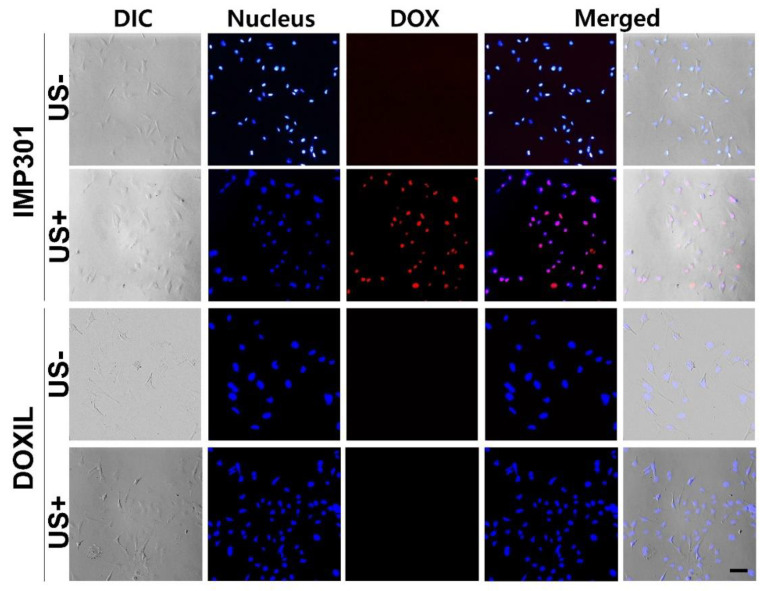
Fluorescence cellular uptake images of doxorubicin encapsulated in IMP301 and DOXIL. Nuclei were stained with DAPI (blue). Released DOX is shown in red, and cell morphology was visualized by differential interference contrast (DIC). Scale bar = 30 µm.

**Figure 5 pharmaceutics-14-01314-f005:**
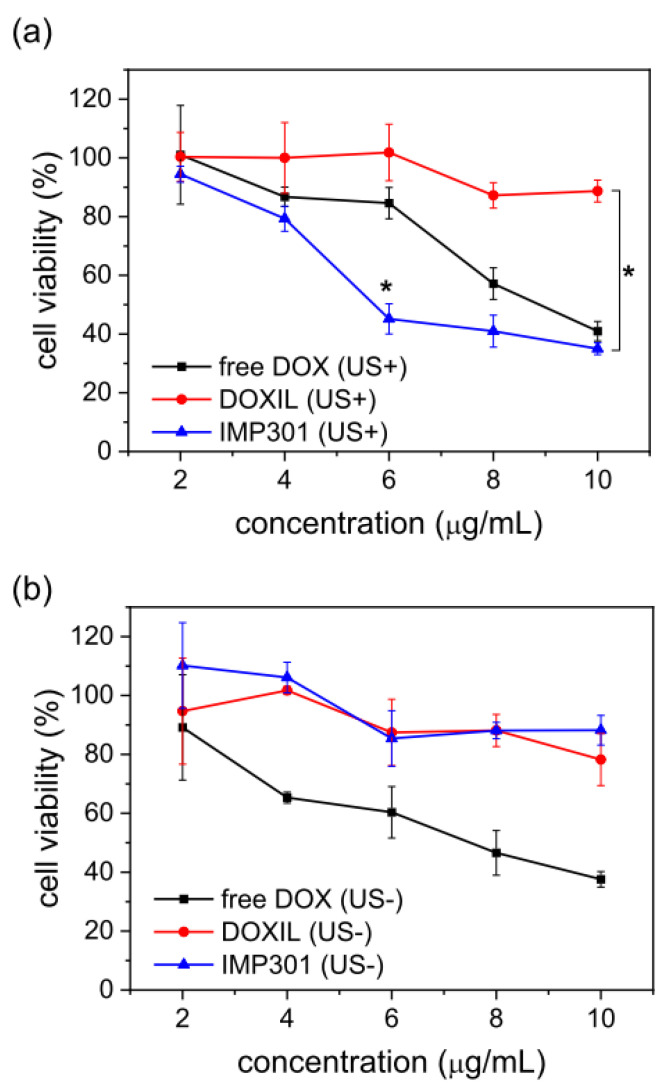
Cytotoxicity of free DOX (black), DOXIL (red), and IMP301 (blue) in MDA-MB-231 cells with (**a**) or without (**b**) ultrasound exposure. * *p* < 0.05.

**Figure 6 pharmaceutics-14-01314-f006:**
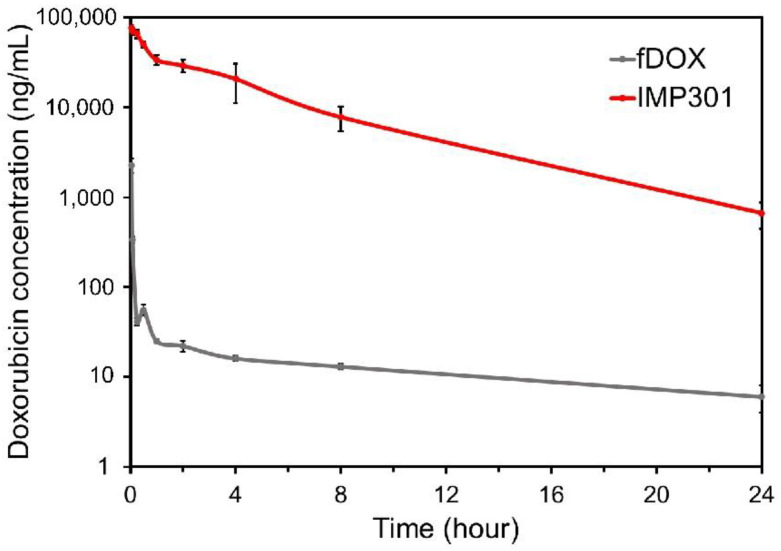
Pharmacokinetic profiles of free doxorubicin (fDOX) and IMP301 in the plasma. The concentrations of fDOX and IMP301 are indicated by the gray and red lines, respectively.

**Figure 7 pharmaceutics-14-01314-f007:**
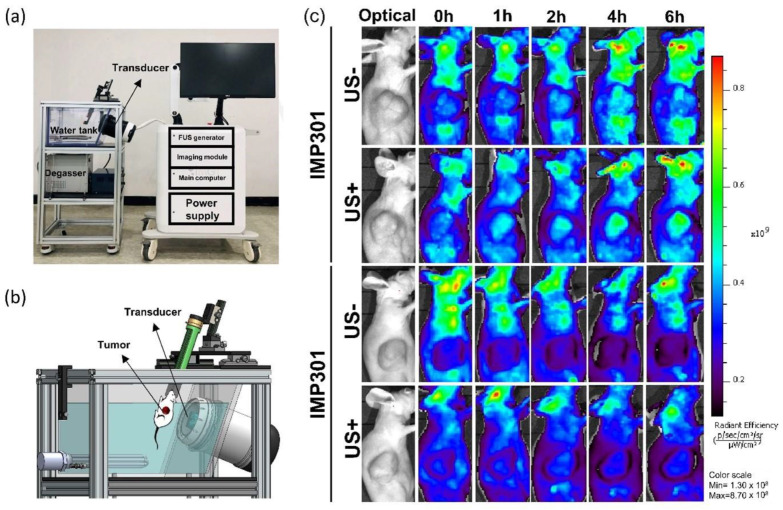
In vivo doxorubicin release from IMP301 and DOXIL under FUS irradiation using IMD-10R. (**a**) System configuration of IMD-10R. (**b**) Schematic illustration of animal setting in water tank of IMD-10R. (**c**) The fluorescence intensity of doxorubicin was specifically detected using spectral unmixing at 408–460 nm excitation and emission for removal of autofluorescence after intravenous injection of DOXIL or IMP301 without or with FUS treatment.

**Figure 8 pharmaceutics-14-01314-f008:**
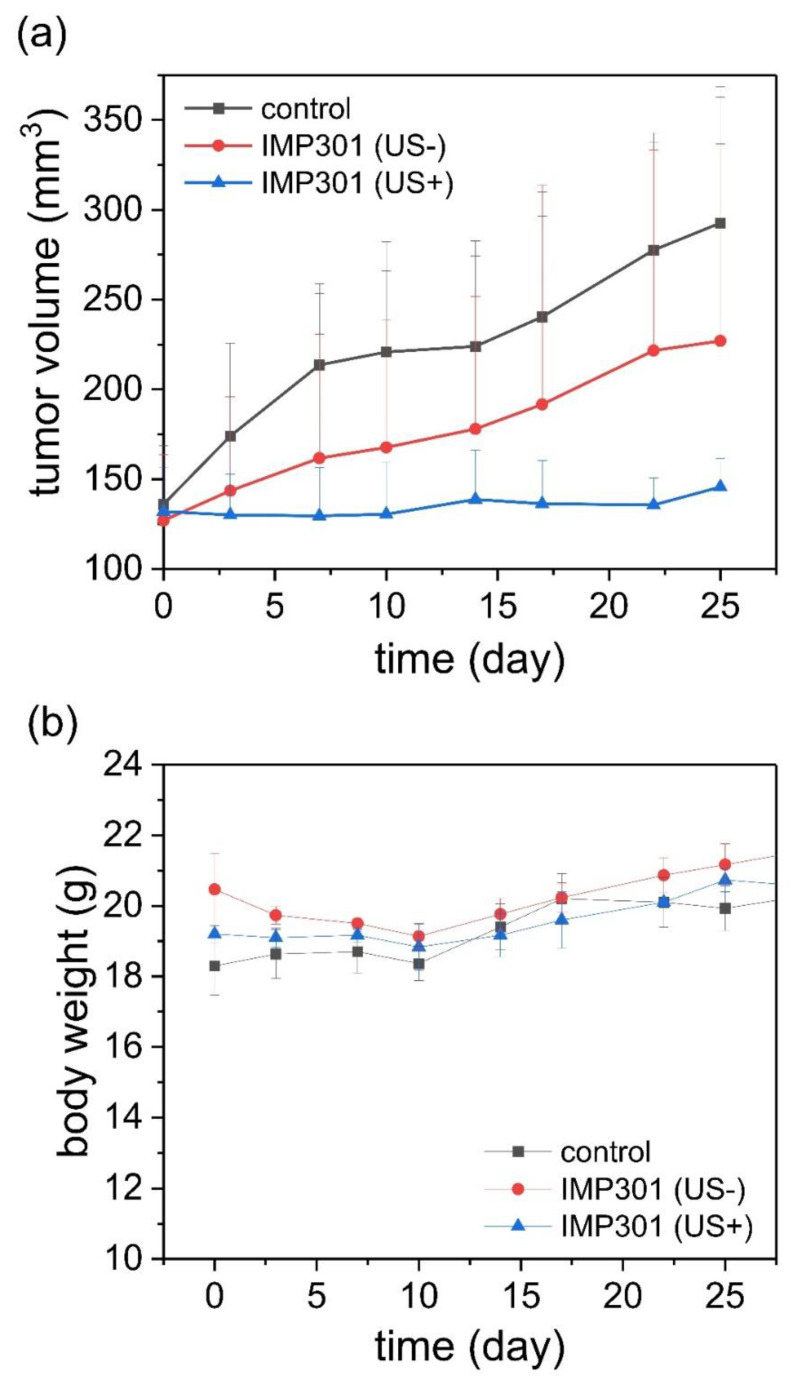
In vivo therapeutic efficacy of the combination of IMP301 and ultrasound in MDA-MB-231 mice after three rounds of treatment. (**a**) Tumor growth in MDA-MB-231 tumor-bearing mice intravenously administrated with saline (control; black line), IMP301 (IMP301 US-; red line), and IMP301 with FUS irradiation (IMP301 US+; blue line). (**b**) Body weight change of mice in each group; (*n* = 3).

**Table 1 pharmaceutics-14-01314-t001:** Doxorubicin content, lipid content, entrapment, size, and PDI of IMP301 during 9 months of storage at 2–8 °C.

Month	DOX (%)	DSPC (%)	DSPC-mPEG2k (%)	Cholesterol (%)	DOPE (%)	MSPC (%)	Entrapment (%)	Size (d·nm)	PDI
**0**	97.5	99.2	106.5	101.5	109.0	108.7	97.8	83.5	0.08
**1**	97.4	96.4	98.1	101.2	109.2	94.7	98.3	86.4	0.10
**3**	96.5	96.5	92.8	96.1	96.1	89.2	96.7	81.2	0.06
**6**	94.4	94.1	93.9	97.1	98.3	91.0	98.6	78.6	0.07
**9**	94.8	94.5	93.2	97.3	98.5	90.0	98.2	83.5	0.12

## Data Availability

Not applicable.

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
