# Peer review of "Ultrasound-Responsive Liposomes for Targeted Drug Delivery Combined with Focused Ultrasound"

_pharmaceutics, 2022, doi:10.3390/pharmaceutics14071314_

Round 1
Reviewer 1 Report
In the proposed paper, titled “Ultrasound-responsive liposomes for targeted drug delivery combined with focused ultrasound”, the authors synthesized and characterized ultrasound-responsive liposomes. They tested them as carriers for the delivery of the anticancer drug doxorubicin, which was able to induce cancer cell death. To this aim, the conducted in vitro and in vivo studies on MDA-MB-231 xenografted mice cells, demonstrating that doxorubicin, loaded into liposomes, was released in response to the ultrasound irradiation. Finally, the authors stated that the combination of ultrasound and liposomes greatly suppressed tumour growth.
The manuscript might be quite interesting and relevant to pharmaceutics, nano-biotech, nanomedicine and, more in general, the medical community, however, it requires several modifications before publication on Pharmaceutics. There are some relevant formal and technical issues that must be addressed.
1. From a formal point of view, the article must be fully revised. In my opinion, the introduction is poorly written: although correct information is reported, it is very poorly presented. For instance, there is no mention of doxorubicin, its pharmacokinetics, etc. and it must be reported.
In addition, the purpose of the work is reported in a few lines at the end of the introduction. It is very difficult to understand the purpose of the manuscript.
2. The manuscript is rich in abbreviations and acronyms, some of which are commonly used in numerous scientific articles (DOX, NIR, MTT…) while others are more specific and less common. Usually, the use of the acronyms should facilitate the reader but, not in this case. For me, it is mandatory to insert a list of acronyms at the beginning of the manuscript.
3. The quality of the images on the paper is, in general, bad. The authors must increase the quality of the images, the size of the curve (Fig. 1 (b) and (d)), the dots (Fig. 5), and all the axis titles/numbers.
4. In section 3.2 the authors state: “The morphology of liposomes changed after ultrasonic irradiation.” How do they quantify this modification in the liposome morphology? What kind of variation was observed? size, shape, circularity? The authors must justify this statement.
5. Why did you choose MDA-MB-231 cells in this research?
6. The images of the cells shown in Fig. 4 are difficult to observe and I cannot distinguish/observe the cells. Authors must provide larger images: in this way, It will be possible to observe the impact of DOX on cell culture. In addition, the authors should clarify why the red fluorescence is the DOX uptake. Lastly, the scale bar is not shown.
7. In my opinion, the impact of DOX on cell morphology must be reported by the authors, especially since confocal images have been added to the manuscript. How does the shape of the cells vary? and the size?
Author Response
Reviewer 1
In the proposed paper, titled “Ultrasound-responsive liposomes for targeted drug delivery combined with focused ultrasound”, the authors synthesized and characterized ultrasound-responsive liposomes. They tested them as carriers for the delivery of the anticancer drug doxorubicin, which was able to induce cancer cell death. To this aim, the conducted in vitro and in vivo studies on MDA-MB-231 xenografted mice cells, demonstrating that doxorubicin, loaded into liposomes, was released in response to the ultrasound irradiation. Finally, the authors stated that the combination of ultrasound and liposomes greatly suppressed tumour growth.
The manuscript might be quite interesting and relevant to pharmaceutics, nano-biotech, nanomedicine and, more in general, the medical community, however, it requires several modifications before publication on Pharmaceutics. There are some relevant formal and technical issues that must be addressed.
- From a formal point of view, the article must be fully revised. In my opinion, the introduction is poorly written: although correct information is reported, it is very poorly presented. For instance, there is no mention of doxorubicin, its pharmacokinetics, etc. and it must be reported.
à I agree reviewer’s comment. The introduction was fully revised following to reviewer’s opinion about pharmacokinetics of doxorubicin and the purpose of this study.
In addition, the purpose of the work is reported in a few lines at the end of the introduction. It is very difficult to understand the purpose of the manuscript.
à As reviewer’s comments, the purpose of this study was logically demonstrated at the end of the introduction section. Briefly, sonosensitive liposome was developed to overcome toxicity of doxorubicin by encapsulation into the liposome and to enhance antitumor effect by local release of doxorubicin under the focused ultrasound.
- The manuscript is rich in abbreviations and acronyms, some of which are commonly used in numerous scientific articles (DOX, NIR, MTT…) while others are more specific and less common. Usually, the use of the acronyms should facilitate the reader but, not in this case. For me, it is mandatory to insert a list of acronyms at the beginning of the manuscript.
à Abbreviations and acronyms are approximately revised to original words for reader to more effectively understand the manuscript. High intensity focused ultrasound (HIFU) was replace to focused ultrasound (FUS) in the manuscript, because the meanings of both words are not totally different. And Abbreviations with low frequency of use were corrected to full words. Common abbreviations and acronyms are remained after revision.
- The quality of the images on the paper is, in general, bad. The authors must increase the quality of the images, the size of the curve (Fig. 1 (b) and (d)), the dots (Fig. 5), and all the axis titles/numbers.
àFollowing to the comments, all figures were considered for clear and accurate description. All lines, dots and words in all figures are enlarged.
- In section 3.2 the authors state: “The morphology of liposomes changed after ultrasonic irradiation.” How do they quantify this modification in the liposome morphology? What kind of variation was observed? size, shape, circularity? The authors must justify this statement.
à I understood the point of reviewer’s comment above. And I recognize that my explanation in section 3.2 was unclear and comfused. the word ‘morphology’ was not adequate to demonstrate the point of section 3.2. The main topic in section 3.2 is doxorubicin release under the focused ultrasound. And the amount of released doxorubicin was quantitively analysed as shown figure 2(a). And The release of doxorubicin was visualized by TEM images. Generally, liposomes including IMP301 encapsulate Doxorubicin in the core space by mechanism of ion exchange. And encapsulated doxorubicin was crystalized by exchange of hydrochloride ion to sulfate ion. Therefore, IMP301 encapsulating doxorubicin figured ellipse coffee bean shape, as shown in figure 2b. On the other hands, IMP301 demonstrated circular particle after release of doxorubicin under the FUS exposure, as shown Figure2c. The crystalized doxorubicin in the core of IMP301 was fully or partially disappeared by FUS exposure.
I agreed the reviewer’s comment and revised section 3.2 in this manuscript for clear understanding.
- Why did you choose MDA-MB-231 cells in this research?
à As reviewer know, MDA-MB-231 cell is triple negative breast cancer cell line. And doxorubicin hydrochloride has been still administrated for the treatment of triple negative breast cancer in clinical field. Therefore, MDA-MB-231 cell was chosen in this research and we expect that IMP301 can be capable of application to treat triple negative cancer in clinical field.
- The images of the cells shown in Fig. 4 are difficult to observe and I cannot distinguish/observe the cells. Authors must provide larger images: in this way, It will be possible to observe the impact of DOX on cell culture. In addition, the authors should clarify why the red fluorescence is the DOX uptake. Lastly, the scale bar is not shown.
à I agreed to reviewer’s comment. Unfortunately, we could not prepare the fluorescence image with high magnification during revision. However, we replaced the fluorescence image optimal fit to distinguish cell. And as reviewer know, anti-cancer mechanism of doxorubicin is well-known as DNA intercalator and protector to duplication. Therefore, red fluorescence intensity of doxorubicin was co-localized to nucleus after cellular uptake. We showed the behaviour of doxorubicin by figure 4. Lastly, scale bar was figured, following to reviewer’s comment.
- In my opinion, the impact of DOX on cell morphology must be reported by the authors, especially since confocal images have been added to the manuscript. How does the shape of the cells vary? and the size?
à I understand the comment about anticancer effect of doxorubicin. This article could be higher quality if the impact of doxorubicin to cell apoptosis was proven by in vitro study. However, the apoptosis by doxorubicin has reported by numerous research. And this study is focused on development of ultrasound-triggered drug delivery system not on biology of doxorubicin. We hope that the reviewer understands our scope of this article. And the size of MDA-MB-231 cell is known as about 20µm. (ref. Sinead Connolly et al, The in vitro inertial positions and viability of cells in suspension under different in vivo flow conditions, Scientific Reports, (2020), 10, 1711.

Reviewer 2 Report
Dear Authors,
The manuscript submitted by Kim & co-workers titled “Ultrasound-responsive liposomes for targeted drug delivery combined with focused ultrasound” is interesting and useful in targeted drug delivery, However in my opinion some suggestion are as below:
1. In Abstract, author need to add with some results, instead writing like a review abstract.
2. What is this IMP301? Is it the code of formulation developed in company?
3. How much tumor growth was suppressed in combination? Need to be mentioned in Abstract last sentence.
4. What is the headspace of 10 mL liposomes formulation packed in a 20 mL vial and why?
5. Why authors stored formulation at 2-8 °C in dark?
6. In Section 2.7, details of DOXIL are required including batch number.
7. In MTT assay author kept 1x104 cell per vial and kept for 72 h, I think the cells population is too high, please clarify.
8. In Table 1, why % entrapment was increased from 97.8 to 98.2 and particle size was same from 83.5 to 83.5 from 0 to 9 months? Please clarify.
9. As we know that the DOX create the cardiotoxicity, author have any data or proof regarding cardiotoxicity of IMP301.
10. Proper literature on liposomes is required with recent 2 years research studies, I am unable to find the references of 2022 and 2021 (2 references cited only)?
Author Response
Reviewer 2
Dear Authors,
The manuscript submitted by Kim & co-workers titled “Ultrasound-responsive liposomes for targeted drug delivery combined with focused ultrasound” is interesting and useful in targeted drug delivery, However in my opinion some suggestion are as below:
- In Abstract, author need to add with some results, instead writing like a review abstract.
à Following to reviewer’s comment, abstract was revised with some result.
- What is this IMP301? Is it the code of formulation developed in company?
à The reviewer was correct. The IMP301 is the code of formulation in our company.
- How much tumor growth was suppressed in combination? Need to be mentioned in Abstract last sentence.
à As followed to reviewer comment, the anticancer efficiency was demonstrated to abstract.
- What is the headspace of 10 mL liposomes formulation packed in a 20 mL vial and why?
à In my private opinion, reviewer probably mentioned about section 2.6, stability test. As reviewer knows, IMP301 has been developed for anticancer therapeutics via intravenous injection. And we have tried to apply for Phase I human trials. Therefore, stability test was investigated using suitable vial. And the 20 mL sterilized vial with aluminium cap has been chosen for submission to Investigational New Drug approval.
- Why authors stored formulation at 2-8 °C in dark?
à The storage condition for doxorubicin hydrochloride is 2-8 °C in dark. And liposome should be also stored at 2-8 °C degree in general. Therefore, we followed regulation of both storage condition.
- In Section 2.7, details of DOXIL are required including batch number.
à Following to the reviewer’s comment, the batch number of DOXIL was demonstrated in the manuscript.
- In MTT assay author kept 1x104cell per vial and kept for 72 h, I think the cells population is too high, please clarify.
à As reviewer know, 1x104 cell per vial is under the range of MTT assay protocol. And MDA-MB-231 slowly proliferates with the doubling time of about 1.3-1.5 day. Therefore, in order to verify anticancer effect by IMP301, MDA-MB-231 cell was incubated for 72h for analysis of cell viability in this study.
- In Table 1, why % entrapment was increased from 97.8 to 98.2 and particle size was same from 83.5 to 83.5 from 0 to 9 months? Please clarify.
à We developed method validation for Quality Control (QC) in all section such as size distribution, entrapment efficiency, quantification of doxorubicin and impurity with each reference standard materials. And all test contents have been set with standard range, which has been investigated by function of IMP301 such as release ratio of doxorubicin by ultrasound. Tiny difference of results will be probably occurred by experimental error. And we have been verified that size and entrapment efficiency has been stably maintained in standard range. Regarding to the verification of standard, we briefly comment in manuscript.
- As we know that the DOX create the cardiotoxicity, author have any data or proof regarding cardiotoxicity of IMP301.
à As reviewer know, DOX create the cardiotoxicity and many drug delivery system including liposome (DOXIL) has been studied to overcome limitation of toxicity by selective delivery and protection to random spread of free doxorubicin. Unfortunately, we could not directly suggest the cardiotoxicity result during revision. However, we compared the population of doxorubicin to the heart and attached this result to figure 8. The population of DOX in IMP301 and DOXIL group was lower than free doxorubicin group. This result was briefly demonstrated in section 3.9
- Proper literature on liposomes is required with recent 2 years research studies, I am unable to find the references of 2022 and 2021 (2 references cited only)?
à As reviewer comment, recent references were cited to the introduction section.

Reviewer 3 Report
Doxorubicin is traditionally used for the treatment of cancer. Among various nanomedicines, liposomes have been the first commercialized drug carriers because of their biocompatibility and versatile characteristics. Doxil is a marketed liposomal doxorubicin, which was compared with developed ultrasound-responsive liposomes containing doxorubicin (IMP301). Ultrasound-induced release of liposomes has been studied using plane-wave ultrasound and high-intensity focused ultrasound (HIFU). In this study, the fabrication and characterization of ultrasound-responsive drug-loaded liposomes, their pharmacokinetics, biodistribution, and in vivo efficacy were investigated.
The concept of the study is interesting and useful. It is very important to increase the effectiveness of anticancer therapy.
Comments and questions:
“Introduction” part:
The study focused on the ultrasound-induced release of liposomes. I have missed the literature review of ultrasound applicability in medicinal therapy. How does it used before? More information about given ultrasound techniques: plane-wave ultrasound and high-intensity focused ultrasound (HIFU) Need references.
“Materials and Methods” part:
In section 2.7
In case of in vitro cellular uptake tests, three groups (IMP301, Doxil and free DOX) were compared with the control group based on the description. But, in the “Results” part only the results of IMP301, Doxil were presented.
Correct the description.
In section 2.8
In case of description of the pharmacokinetics investigation, three groups (free DOX, IMP301 and Doxil) are mentioned but, in the “Results” part only the results of IMP301 and free DOX were presented.
Correct the description.
“Result and Discussion” part:
In section 3.1
The entrapment efficiency or loading efficiency one of the most important property of the liposomes. Which was used? Define it with equation. How do you calculate it? And in section 3.1 the data is 97.6% and in section 3.4 the data is 97.8%
Clear these part of the measurements.
In my opinion, the presentation of release behaviour of liposomes vs US exposure time is more fitted to section 3.2. or 3.3.
In section 3.4
In case of long term stability I have missed the statistical description. How many parallel measurements were investigated, and which were the SD values?
Improve the section.
In section 3.5
What does the “DIC” abbreviation means?
Give the explanation.
In section 3.8
Authors write “IMP301 and DOX ultrasound –triggered release was compared…..”
But I think “IMP301 and DOXIL ………” is the right.
Correct it.
Author Response
Reviewer 3
Doxorubicin is traditionally used for the treatment of cancer. Among various nanomedicines, liposomes have been the first commercialized drug carriers because of their biocompatibility and versatile characteristics. Doxil is a marketed liposomal doxorubicin, which was compared with developed ultrasound-responsive liposomes containing doxorubicin (IMP301). Ultrasound-induced release of liposomes has been studied using plane-wave ultrasound and high-intensity focused ultrasound (HIFU). In this study, the fabrication and characterization of ultrasound-responsive drug-loaded liposomes, their pharmacokinetics, biodistribution, and in vivo efficacy were investigated.
The concept of the study is interesting and useful. It is very important to increase the effectiveness of anticancer therapy.
Comments and questions:
“Introduction” part:
The study focused on the ultrasound-induced release of liposomes. I have missed the literature review of ultrasound applicability in medicinal therapy. How does it used before? More information about given ultrasound techniques: plane-wave ultrasound and high-intensity focused ultrasound (HIFU) Need references.
à As reviewer’s comment, introduction was fully revised. And HIFU application and ultrasound techniques was involved.
“Materials and Methods” part:
In section 2.7
In case of in vitro cellular uptake tests, three groups (IMP301, Doxil and free DOX) were compared with the control group based on the description. But, in the “Results” part only the results of IMP301, Doxil were presented.
Correct the description.
à In previous manuscript, control group was not included because we tried to emphasize comparison of ultrasound sensitivity between IMP301 and DOXIL. However, free DOX group was added in revision manuscript, as followed reviewer’s comment. And all figures were revised for accurate description.
In section 2.8
In case of description of the pharmacokinetics investigation, three groups (free DOX, IMP301 and Doxil) are mentioned but, in the “Results” part only the results of IMP301 and free DOX were presented.
à We appreciate the reviewer’s correction. The mistake was induced because IMP301, DOXIL and free doxorubicin were frequently compared in this manuscript. We correct the description of pharmacokinetics in revised manuscript.
Correct the description.
“Result and Discussion” part:
In section 3.1
The entrapment efficiency or loading efficiency one of the most important property of the liposomes. Which was used? Define it with equation. How do you calculate it? And in section 3.1 the data is 97.6% and in section 3.4 the data is 97.8%
Clear these part of the measurements.
à As followed to reviewer’s comment, the analytical method for entrapment efficiency was added to revision manuscript. Briefly, PD-10 column was used to trap unencapsulated doxorubicin for purification of IMP301. And purified IMP301 encapsulating doxorubicin was quantified by absorbance at 475 nm. And difference encapsulation efficiency between section 3.1 and 3.4 was from Batch No. To protect misunderstanding, we revised entrapment efficiency with standard deviation. (97.1 ± 1.44 %)
In my opinion, the presentation of release behaviour of liposomes vs US exposure time is more fitted to section 3.2. or 3.3.
à Following to the reviewer’s comment, Figure 1 in section 3.2 was revised.
In section 3.4
In case of long term stability I have missed the statistical description. How many parallel measurements were investigated, and which were the SD values?
Improve the section.
à I agreed to reviewer’s comment. Unfortunately, long term stability was investigated by 1 time at each point. In fact, there were many results for long term stability such as impurity, osmolality and residual organic solvent for Quality Control of IMP301. Therefore, we analysed mixture of IMP301 vials(N=3) for convenience, because many QC items should be measured.
In section 3.5
What does the “DIC” abbreviation means?
Give the explanation.
à DIC abbreviation means differential interference contrast. As reviewer mentioned, explanation of DIC was involved in manuscript
In section 3.8
Authors write “IMP301 and DOX ultrasound –triggered release was compared…..”
But I think “IMP301 and DOXIL ………” is the right.
Correct it.
à We appreciate the reviewer’s correction. The mistake was induced because IMP301, DOXIL and free doxorubicin were frequently compared in this manuscript. We correct the description in revised manuscript.

Reviewer 4 Report
The work is well written and the result are presented clearly. Just some details to be improved and/or corrected
-Pag 2 Introduction at the end when refer to bubbles "resonant size and then explode...
bubbles always IMPLODE because of ultrasound effect
-pag 6 figure 1 b color of the curves should be changed for better clarity ( black and red for example)
-pag 6-7 paragraph 3.2-3.3 there isn't a report of the temperature of the medium during US trials for DOX release. A value of temperature should be reported somewhere considering all the different ultrasonic test ( PRF, duty cycle and so on).
Author Response
Reviewer 4
The work is well written and the result are presented clearly. Just some details to be improved and/or corrected
-Pag 2 Introduction at the end when refer to bubbles "resonant size and then explode...
bubbles always IMPLODE because of ultrasound effect
à As reviewer mentioned, ‘explode’ was changed to ‘implode’.
-pag 6 figure 1 b color of the curves should be changed for better clarity ( black and red for example)
à As reviewer mentioned, figure 1 b color of the curves was changed. And all figures were revised for better clarity.
-pag 6-7 paragraph 3.2-3.3 there isn't a report of the temperature of the medium during US trials for DOX release. A value of temperature should be reported somewhere considering all the different ultrasonic test ( PRF, duty cycle and so on).
à As reviewer mentioned, influence of temperature for release of doxorubicin was demonstrated in revised manuscript. And supporting information involved release behavior of doxorubicin release on temperature and simulation result of temperature on optimal ultrasound condition.

Round 2
Reviewer 1 Report
Dear Authors,
I appreciate the answers that you gave me and all the modifications in the manuscript. I believe that the quality and readability of the manuscript have improved considerably. The images added in the text, although not eye-catching, help readers understand the article.
Reviewer 2 Report
The revised version manuscript is acceptable
Reviewer 3 Report
The corrected version is acceptable.